# Azides in the Synthesis of Various Heterocycles

**DOI:** 10.3390/molecules27123716

**Published:** 2022-06-09

**Authors:** AbdElAziz A. Nayl, Ashraf A. Aly, Wael A. A. Arafa, Ismail M. Ahmed, Ahmed I. Abd-Elhamid, Esmail M. El-Fakharany, Mohamed A. Abdelgawad, Hendawy N. Tawfeek, Stefan Bräse

**Affiliations:** 1Department of Chemistry, College of Science, Jouf University, Sakaka 72341, Al Jouf, Saudi Arabia or aanayl@yahoo.com (A.A.N.); waarafa@ju.edu.sa (W.A.A.A.); imibrahim@ju.edu.sa (I.M.A.); 2Chemistry Department, Faculty of Science, Organic Division, Minia University, El-Minia 61519, Egypt; hendawy1976@yahoo.com; 3Composites and Nanostructured Materials Research Department, Advanced Technology and New Materials Research Institute, City of Scientific Research and Technological Applications (SRTA-City), Alexandria 21934, Egypt; ahm_ch_ibr@yahoo.com; 4Protein Research Department, Genetic Engineering and Biotechnology Research Institute GEBRI, City of Scientific Research and Technological Applications (SRTA City), New Borg Al-Arab, Alexandria 21934, Egypt; eelfakharany@srtacity.sci.eg; 5Department of Pharmaceutical Chemistry, College of Pharmacy, Jouf University, Sakaka 72341, Al Jouf, Saudi Arabia; mhmdgwd@ju.edu.sa; 6Institute of Organic Chemistry, Karlsruhe Institute of Technology, 76131 Karlsruhe, Germany; 7Institute of Biological and Chemical Systems (IBCS-FMS), Karlsruhe Institute of Technology, Ggenstein-Leopoldshafen, 76344 Karlsruhe, Germany

**Keywords:** organic azides, click reaction, catalysis, five membered rings, six membered rings, organo-metal heterocycles

## Abstract

In this review, we focus on some interesting and recent examples of various applications of organic azides such as their intermolecular or intramolecular, under thermal, catalyzed, or noncatalyzed reaction conditions. The aforementioned reactions in the aim to prepare basic five-, six-, organometallic heterocyclic-membered systems and/or their fused analogs. This review article also provides a report on the developed methods describing the synthesis of various heterocycles from organic azides, especially those reported in recent papers (till 2020). At the outset, this review groups the synthetic methods of organic azides into different categories. Secondly, the review deals with the functionality of the azido group in chemical reactions. This is followed by a major section on the following: (1) the synthetic tools of various heterocycles from the corresponding organic azides by one-pot domino reaction; (2) the utility of the chosen catalysts in the chemoselectivity favoring C−H and C-N bonds; (3) one-pot procedures (i.e., Ugi four-component reaction); (4) nucleophilic addition, such as Aza-Michael addition; (5) cycloaddition reactions, such as [3+2] cycloaddition; (6) mixed addition/cyclization/oxygen; and (7) insertion reaction of C-H amination. The review also includes the synthetic procedures of fused heterocycles, such as quinazoline derivatives and organometal heterocycles (i.e., phosphorus-, boron- and aluminum-containing heterocycles). Due to many references that have dealt with the reactions of azides in heterocyclic synthesis (currently more than 32,000), we selected according to generality and timeliness. This is considered a recent review that focuses on selected interesting examples of various heterocycles from the mechanistic aspects of organic azides.

## 1. Introduction

Organic azides are organic compounds containing the azide (N_3_) functional group. Due to the hazards associated with their use, few azides are commercially used, although they display interesting applications in organic chemistry. Organic azides have four mesomeric structures (**1a**–**1d**, Figure 1), and their structure is also described as isoelectronic with carbon dioxide. 

The polar character of the azido group has a remarkable effect on their bond lengths and angles. In methyl azide, as an example, the angles of CH_3_-N^1^–N^2^N^3^ and CH_3_N^1^–N^2^–N^3^ are approximately 115.28 and 172.58 Å, respectively [1]. Aromatic azides show slightly shorter bond lengths between N^2^ and N^3^ [1]. Accordingly, an almost linear azide structure is present, with sp^2^ hybridization at N^1^. The polar resonance structures **Ib**–**Id** illustrated that strong IR absorption by a band at nearly 2114 cm^−1^ (phenyl azide [2]). Alkyl azides show absorption in the UV region at 287 nm and 216 nm [2]. They also exhibit weak dipole moment (1.44 *D* for phenyl azide) [2]. Azido group in aromatic substitution reactions directs to *ortho*- and *para*-positions. 

Organic azides engage in useful organic reactions, as the terminal nitrogen is mildly nucleophilic. Generally, nucleophiles attack the azide at the terminal nitrogen N_γ_, while electrophiles react at the internal atom N_α_ [3]. Azides easily extrude diatomic nitrogen, a tendency that is engaged in many reactions, such as the Staudinger ligation or the Curtius rearrangement [4]. Azides can be reduced to amines by hydrogenolysis [5] or with a phosphine (e.g., triphenylphosphine) in the Staudinger reaction [5]. Organic azides can react with phosphines to give iminophosphoranes, which can be hydrolyzed into primary amines (the Staudinger reaction) [6]. They react with carbonyl compounds to give imines (the aza-Wittig reaction) [7,8] or undergo other transformations. Thermal decomposition of azides gives nitrenes, which participate in various reactions; vinyl azides decompose into 2*H*-azirines [9]. 

Since organic azides are highly reactive and have been long established as versatile building blocks in assembling structurally diverse *N*-containing heterocycles, converting organic azides into high-value compounds, such as heterocycles, would be greatly valued and a subject of enormous current interest. Currently, well over 32,000 total and nearly 1000 in 2021 showed interest in this type of chemistry.

## 2. Some Synthetic Procedures of Organic Azides

### 2.1. From Diazonium Salts

The aryl diazonium salts were decomposed readily on reacting with azide ions (NaN_3_ or Me_3_SiN_3_) to the corresponding aryl azide without a catalyst. As an example, the facile conversion of 5-amino-2-(2,6-dioxopiperidin-3-yl)isoindoline-1,3-dione (**1**) into 5-azido-2-(2,6-dioxopiperidin-3-yl)isoindoline-1,3-dione (**2**) via a two-step reaction involving diazotization followed by azidation using sodium azide as a precursor of azide ion [10] (Figure 1). 

### 2.2. Via S_N_Ar Reactions (Nucleophilic Aromatic Substitution Reactions)

As an example, synthesis of 2-azido-3-nitropyridines (**4**) from 2-chloro-3-nitropyridines (**3**) using NaN_3_ as the source of nucleophile (Figure 2) was established as shown in Figure 2 [11,12]. 

### 2.3. From Lithium-Reagent

The reaction of aromatic halides **5** with lithium reagent (*t*-BuLi) followed by the reaction with tosyl azide gave the corresponding aryl azide **6** in 96% yield (Figure 3) [13]. 

### 2.4. From Aryl Hydrazines

Kim et al. [14] reported the synthesis of aromatic azide **8** from the reaction of arylhydrazine **7** with nitrosyl ion (Figure 4). 

## 3. Chemistry of Azides

### 3.1. Azide as Aminating Group

#### 3.1.1. Synthesis of 8-Aminoquinoline

The biologically active 8-aminoquinoline **10** was obtained through Ir(III)-catalyzed C8-amination of C2-selenylated quinoline *N*-oxide **9** with tosyl azide [15] (Figure 5).

#### 3.1.2. Synthesis of Amino Furo/Pyrroloindole Derivatives

Zhang and others [16] reported that tryptophols or tryptamines reacted with aryl azides to produce 3a-nitrogenous furoindolines and pyrroloindolines 3a-nitrogenous indole alkaloids. Using a nitrogen source, the reaction proceeded via nitrene transfer/cyclization under copper-catalyzed conditions. Firstly, the reaction was investigated to stand at the optimal reaction conditions indicated in Figure 6. Starting with tryptophol (2-(1*H*-indol-3-yl)ethanol) (**11a**) and 1-azido-4-methoxybenzene, the corresponding furoindole **12a** was obtained (Figure 6). The investigation revealed that the conditions mentioned in entry 14 were chosen to be the optimal reaction conditions for all substrates (CuBH_4_(PPh_3_)_2_ + **L7** (12 mol%), DCE 0.5 h). 

Utilizing by the aforementioned optimized condition, a variety of tryptophols (**11a**–**n**) and tryptamine substrates (**11o**–**r**) were taken to react with 1-azido-4-methoxybenzene, as shown in Figure 7. According to the electronic nature or positions of the substituents, the reactions proceeded smoothly to give the target products **12a**–**r** in high yields ranging from 72 to 99%. Moreover, different aryl azides were selected to react with tryptophol (2-(5-chlorobenzofuran-3-yl)ethanol) (**11e**) under the optimized reaction conditions to investigate the effect of substituents on the products’ yields.

Another series of compound **12** was prepared, with azides having electron-donating and -withdrawing groups. The reaction proceeded smoothly to give the corresponding products **12s**–**f`** in moderate to excellent yields [16] (Figure 8). 

The suggested mechanism describes the formation of furroindole **12a**, as shown in Figure 9 [16]. Firstly, the azide moiety reacted with copper complex to produce copper nitrene complex **A**, which abstracts a hydrogen atom from compound **11a** to form the copper aminyl species **B** and imine radical **C**, which combines to produce the intermediate **D**. The catalyst moiety was then excluded from the intermediate **D** to form the imine species **E**. Finally, **E** was converted to the desired product **12a** via an intramolecular nucleophilic addition [16].

### 3.2. Azidation

#### Synthesis of 3-Azido-Tetralins, Chromanes, and -Tetrahydroquinolines

Porter et al. [17] reported the stereoselective synthesis of 3-azido-tetralins, chromanes, and -tetrahydroquinolines via a tandem allylic azide rearrangement/Friedel-Crafts alkylation. The allylic azides **13a**–**f** were cyclized to the corresponding tetralines **14a**–**f** in 58–88% (Figure 10). 

In continuation of the optimized procedure [17], the ethereal allylic azides **15a**–**i** were converted into chromanes **16a**–**i** in 34–97% yields (Figure 11). 

On the other hand, aniline-derived allylic azides **17a**–**f** carrying the *N*-protecting group were also cyclized using the tandem process to tetrahydroquinolines **18a**–**f** in good yields of 57–81% [17] (Figure 12). 

Finally, the tetraline **14a** was converted into pyrrolidine **19** [17]. At the same time, the cycloaddition reaction of the tetraline **14a** with dimethyl acetylene dicarboxylate gave the triazole **20** (Figure 13). 

An interest application of previously reported work, was directed towards synthesizing a large family of botanical natural products group named husbanan [17]. Husbanan was synthesized from ethyl 2-phenethylcyclohex-1-enecarboxylate (**21**), which initially underwent reduction followed by partial re-oxidation to the aldehyde **22** (i.e., from ester to alcohol then aldehyde using tetrapropylammonium perruthenate (TPAP), and *N*-methylmorpholine *N*-oxide (NMO)). Aldehyde **22** was elaborated by Corey-Chaykovsky epoxidation. Epoxide **23** was opened with NaN_3_ in acetone/water yielding the allylic azide **24**. Imidate **25** was isolated after activation with trichloroacetonitrile. Finally, reduction of imidate **25** gave **26** on the presence of dicyclohexyl borane ring closer to the hasubanan product **27** [17] (Figure 14).

## 4. Organic Azides in the Synthesis of Heterocycles

Organic azides have synthesized various heterocycles of the five-member ring with one heteroatom, such as pyrroles. They are also involved in synthesizing heterocycles with two heteroatoms, such as pyrazole and isoxazole, oxazole, thiazole, oxazine, and pyrimidine. In addition, heterocycles containing three heteroatoms, such as 1,2,3-triazoles and thiadiazole, are also included. Furthermore, organic azides are used in synthesizing heterocycles of six-membered rings and with one heteroatom, such as pyridine, isoquinoline, and phenanthridine. Heterocycles with four heteroatoms, such as tetrazole, are also discussed. Synthetic interest applications of organometal heterocycles (i.e., phosphorus-, boron-, and aluminum-containing heterocycles) were also investigated. Figure 2 indicates the contribution of organic azides in heterocyclic synthesis.

### 4.1. Synthesis of the Pyrrole Ring

Dong and others [18] reported that dipyrrin-supported nickel catalyst (^AdF^L)Ni(py) (^AdF^L: 1,9-di(1-adamantyl)-5-perfluorophenyldipyrrin; py: pyridine) catalyzed the productive intramolecular C−H bond amination to give *N-*heterocyclic products **28a**–**k** using aliphatic azide substrates. The catalytic amination conditions were mild, requiring 0.1−2 mol% catalyst, and occurred at room temperature. The amination process occurred using different substrates having multiple activatable C−H bonds (Figure 15 and Figure 16). The selective catalyst showed high chemoselectivity favoring C−H bonds in ethers, halides, thioethers, esters, etc. (Figure 17). Sequential cyclization of substrates with ester groups was achieved to provide facile preparation of indolizidine derivatives found in various alkaloids [18]. 

The amination cyclization reaction mechanism is illustrated in Figure 18. Benzene (4-azido-4-methyl pentyl), as an example, releases pyridine and N_2_ from **L** to give the corresponding nickel iminyl species **A**. The next step would be a hydrogen atom abstraction to give radical **B**, followed by radical recombination to give the cyclized product **28a** [18].

Sun et al. [19] have reported the diastereoselective synthesis of Boc-protected 4-methylproline carboxylates **35**, starting with the reduction of the azido compound **29**. Selective cleavage of the primary TBS-alcohol **30** was performed using NH_4_F in MeOH at room temperature for 6 h. Oxidation of the alcohol **30** was achieved using 2-iodoxybenzoic acid (IBX) in DMSO at 30 °C, and the resulting aldehyde **31** was subsequently transformed into the corresponding methyl ester **32** by adding KOH/I_2_/MeOH. Deprotection of ester **32** with camphorsulfonic acid (CSA) afforded the corresponding alcohol **33**. Tosylation of alcohol **33** with *p*-toluenesulfonyl chloride in the presence of 1,4-diazabicyclo[2.2.2]octane (DABCO) as a base, gave compound **34** in high yield (90%). Finally, the two-step azide reduction/intramolecular S_N_2 cyclization procedure was performed to obtain the desired Boc-protected (2*S*,4*S*)-4-methylproline carboxylate (**35**) in 90% yield (Figure 19) [19].

Using chiral bisoxazoline-copper (BOX-Cu) complexes as catalysts, the cyclization process of asymmetric azide-ynamides via *α*-imino copper carbene intermediates polycyclic *N*-heterocycles with high enantioselectivities (up to 98:2 e.r) was performed [20]. *N*-Styryl benzyl-tethered(azido)ynamide(*N*-((2-(azidomethyl)phenyl)ethynyl)-*N*-cinnamyl-4-methylbenzene sulfonamide) (**36a**) was selected as the model substrate, and Cu(CH_3_CN)_4_BF_4_ was used as a catalyst to give ((8b*R*,9*R*,9a*S*)-9-phenyl-2-tosyl-2,4,9,9a-tetrahydro-1*H*-benzo[*e*]cyclopropa[*c*]indole) (**37a**) in 86% yield and *N*-cinnamyl-4-methyl-*N*-(naphthalen-2-yl)benzenesulfonamide (**38**) as a side product (Figure 20). The reaction was applied to various *N*-(azido)ynamide **36a**–**z**. Firstly, different selected *N*-protecting groups of the ynamides **36a**–**e** were examined, and the reaction proceeded smoothly when Ts-,Bs-, Ns-, SO_2_Ph-, and Ms- were used as protecting groups (PG = protecting group, Bs = 4-bromobenzene-sulfonyl, Ns = 4-nitrobenzene-sulfonyl)(azido)ynamides) producing the desired tetracyclic heterocycles **37a**–**e** in 63–83% yields. In addition, various aryl-substituted benzyl-tethered (azido)ynamides **36f**–**m,** having either electron-withdrawing and/or electron-donating groups, were also examined, and the corresponding cyclopropanation products **37f**–**m** were obtained in good yields. The reaction was also applied to the thienyl- and furanyl-substituted (azido)ynamides **36n,o** to give **37n** and **37o** in 76% and 67% yields, respectively. Different substituents on the phenyl ring **36p**–**v** (F, Cl, Br, Me, and OMe) were also examined, and products **37p**–**v** were obtained in 63–88% yields. Piperidine fused tetracyclic heterocycle **37w** was also obtained in 71% yield. Moreover, methyl-, ethyl-, and even dimethyl-substituted *N*-allyl (azido)ynamides **36x**–**z** were also suitable substrates for this type of cyclization to give products **37x**–**z** in 69–86% yields (Figure 21). In addition, the reaction was extended to investigate the copper-catalyzed cyclization for *N*-propargyl benzyl-tethered (azido)-ynamides **39a**–**u** (Figure 22) to synthesize 3*H*-pyrrolo[2,3-*c*]isoquinolines **40a**–**u** using 10 mol% of Cu(CH_3_CN)_4_PF_6_ as the catalyst and 2 equiv. of DDQ (4,5-dichloro-3,6-dioxocyclohexa-1,4-diene-1,2-dicarbonitrile) as oxidant [20].

Under thermal and UV light exposure, vinyl azides have been known to decompose into nitrenes and/or 2*H*-azirines, and they have been widely utilized to synthesize various *N*-heterocycles [21]. A photocatalyst-free visible light synthesized substituted pyrroles **42a**–**i** from *α*-keto vinyl azides **41a**–**i**. The optimized reaction condition was determined by applying the reaction on compound **41a**, and it was observed that the optimal reaction condition was irradiation of 0.1 M solution of **41a** in DCE with a blue LED (7 W) light (Figure 23 and Figure 24).

The reaction mechanism was described as a result of the denitrogenative photodecomposition process of *α*-keto vinyl azides, 1,3-amino group migration, and coupling of intermediates **43**–**48** with secondary amines, as shown in Figure 25 [22].

### 4.2. Synthesis of the Pyrazole Ring

Quiclet-Sire et al. [23] reported syntheses of tetrahydropyrrolo-pyrazole from hydrazones of pendant alkene using iodine in a basic medium as a catalyst. In contrast, Jahn et al. [24] demonstrated the construction of tetrahedropyrrolo-pyrazole **46**, **47a**–**m** via Aza-Michael addition/[3+2]cycloaddition between *α*,*β*-unsaturated esters **43a**–**c** or amide **43d** and allylic amines **44a**–**f** using nonaflyl azides **45** ((F_3_C(CF_2_)_3_SO_2_N_3_, NfN_3_), acting as aza- transfer reagent) in the presence of *n*-BuLi. The reaction proceeded regioselectively with more than 12:1 1,8-*trans*/1,8-*cis* selectivity (Figure 26) [24].

The mechanism for this reaction is illustrated in Figure 27. Initially, lithiation of the amine **44** gave the lithium amide **48**, which coordinates to the carbonyl group of **43**, followed by transfer of the amide group to the *β*-position via the formation of intermediate **49**. (*Z*)-Enolate then **49** couples with nonaflyl azide (NfN_3_) to form the unstable triazenide **50**. Protonation **50** would give the intermediate **51,** followed by the formation of diazo intermediate **52**. Finally, diastereoselective cycloaddition step takes place via transition state **53** and **54** to give the diastereoisomers **46** (1,8-*trans*) and **47** (1,8-*cis*) (Figure 27) [24].

Moreover, Just and others [25] reported the catalytic syntheses of fused tricyclic pyrrolidinopyrazolines via aza-Michael cycloaddition of cyclic amines **55** as well as (*R*)-*N*-benzyl-cycloalkenyl amines **58** with NfN_3_ **45**; the reaction was catalyzed by lithium chloride (LiCl). Diastereoselective products have been obtained in good yields (68–84%) for the tricyclic products (*trans*)-**56a**–**f** and (*cis*)-**57a**–**f**, while in the case of 5,5,5-tricyclic (*trans*)-**59a**–**d** and (*cis*)-**60a**–**d,** the yield ranged from 72 to 85%. Regarding the optimized reaction conditions (Figure 28 and Figure 29), it was observed that the diastereomers’ yields depend on the nature of cycloalkenylmethyl amines having five- or six-membered rings and *α*,*β*-unsaturated esters bearing alkyl or aryl substituents in *β*-position. In Figure 30, the proposed mechanism for forming the tetrahedral-pyrrolo-pyrazole from the reaction of cycloalkenyl amines and *α*,*β*-unsaturated esters via intermediates **61**–**65** was postulated [25].

Previously, a cycloaddition reaction was reported between cinnamyl azide and methyl acrylates to obtain the tetrahydro-pyrrole-pyrazole [26,27]. Recently, Carlson et al. [28] developed the previously mentioned procedure via stereoselective interaction between allylic azides and acrylates in high yields. The development includes (i) secondary and tertiary azides, (ii) the use of an enantioenriched azide, (iii) cinnamyl azides substituted at the *α* or *β*-carbon, (iv) derivatization of the products, and (v) additional Michael acceptors. Interestingly, it was found the following sequences: (A) the reaction was not completed during the reaction with cinnamoyl azide **66a**, (B) 1 equivalent of acrylate **43** did not consume the azide at room temperature for three d, and (C) quantitative intermediates **67a**–**69a** were obtained (Figure 31). Optimization of acrylates **43** with cinnamoyl azides with aryl group of electrons withdrawing character has been investigated in Figure 32. Re-optimization of reaction conditions such as (i) solvent, (ii) concentration, (iii) temperature, (iv) equivalents of acrylate, (v) time, and (vi) addition of DIPEA. It was found that the reaction proceeds well with a variety of cinnamoyl azides and the yields were improved. When DIPEA was used as a solvent compound, **70b** was obtained with a 94% yield (Figure 33). Optimization of the reaction in the scope of the substrate, incorporating methyl or phenyl group adjacent to the azide, for compound **70o** a diastereomer was observed. Furthermore, cyclic azide resulted in the formation of tricyclic compounds **70u**–**w,** as demonstrated in Figure 34 [28].

### 4.3. Synthesis of Heterocycles Containing Two Heteroatoms

Vinyl azides **71** reacted with trifluoroacetic anhydride **72** in the presence of NEt_3_ to give 5-(trifluoromethyl)isoxazoles **73a**–**as** via denitrogenative cyclization processes (Figure 35) [29].

### 4.4. Synthesis of Oxazole, Thiazole, and Oxazine Derivatives

The reaction of substituted vinyl azides **74** with a combination of substoichiometric amounts of iron(II) chloride and Togni’s trifluomethylating reagent **75** resulted in the formation of 2,2,2-trifluoroethyl-substituted 3-oxazolines **76**, 3-thiazolines **77**, and 5,6-dihydro-2*H*-1,3-oxazines **78**. It was found that the optimal reaction conditions clarified that DCM, DCE, DMF, and 1,4-dioxane were solvents of choice, and the temperature varied from 80°C to ambient temperature (Figure 36) [30].

The proposed mechanism described the formation of compound **76a**. It showed the role of Fe^II^ in the reaction steps and its activation of Togni’s reagent via the formation of intermediates **A**–**C**; deprotonation was then achieved by the Fe^III^ iodobenzoate complex (**D**) to give compound **76a** and iodobenzoic acid **79** (Figure 37) [30].

### 4.5. Synthesis of the Triazole Ring

Reactivity of azides **80** in [3 + 2] cycloaddition with aromatic or aliphatic terminal alkynes **81a**–**i** clarified that 5 mol% of CuMeSal (copper(I) 3-methylsalicylate), with azidotrifluoromethane and other azidoperfluoroalkanes, afforded a range of *N*-bromo tetrafluoroethyl-substituted 1,2,3-triazoles **82a**–**i** in good to high yields (Figure 38). Since the reaction gave only one regioisomer, it was described as highly efficient and regiospecific (the reaction exclusively afforded only the 1,4-disubstetuted triazole derivatives) at room temperature [31].

Ionic liquid/iron(III) chloride as a homogeneous catalyst was applied in the synthesis of 1 1,2,3-triazoles **84a**–**n** from the reaction of substituted azides and substituted styrenes **83a**–**g** [32] (Figure 39).

Regioselective synthesis of 1,4,5-trisubstituted-1,2,3-triazoles **86a**–**p** from the catalyzed reaction between enaminones **85** and aryl azides using 1-methyl pyridinium trifluoromethanesulfonate [mPy]OTf as the ionic liquid in basic medium via 1,3-dipolar cycloaddition (Figure 40). Herein, the reaction selectively generates only the 1,5-disubstituted triazoles as the only possible product over 1,4-disubstituted triazoles. As illustrated in Figure 41, the proposed mechanism demonstrated that the formation of triazoles **86a**–**n** from the reaction of the aryl azides and enaminones **85a**–**g** was taken via the formation of intermediate **87a**–**n** [33]. The reaction was described as retro-Michael addition to give two regioisomers. Elimination of aniline from the two regiosomers afforded the corresponding triazoles **86a**–**n**. Moreover, the reaction took place with complete regioselectivity yielding the regioisomer with the electron-deficient group of the enaminone in position 4 and the alkyl substituent in position 5 as the only product of the reaction.

Zhang et al. [34] reporoom temperatureed the one-pot multicomponent reaction for the syntheses of 5-thiotriazoles **89a**–**u**, **91a**–**m,** and 5-selenotriazole **92a**–**l** scaffolds using sulfur and selenium elements. Firstly, the reaction was displayed between methyl propiolate, benzyl bromide, S_8_, and 4-methoxybenzyl azide (PMBN_3_) and was selected to optimize the reaction conditions. It was clear that the optimized condition was achieved for **89a** in the following conditions: CuI (1.3 equiv), K_2_CO_3_ (2.0 equiv), and S_8_ (3.0 equiv) in MeCN (or DMF), first at 0 °C for 1 h and then at 50 °C for 10 h (Figure 42). The yield was increased with an increasing amount of CuI to 1.3 equivalent and at 50 °C using DMF or MeCN as a solvent. Accordingly, the yields of compounds **89a**–**u** became good compared with the previous method, as shown in Figure 43.

Next, the influence of the alkynes and azides was examined using DMF as the solvent at temperatures ranging from room temperature to 70 °C for the generation of the sulfenylating agent (Figure 44). Aromatic alkynes with methyl, methoxy, and nitro groups on the benzene ring worked well to produce the corresponding 5-thiotriazoles **91a**–**i**. The efficiency of acylacetylenes was demonstrated by the generation of **91j** bearing a reactive hydroxyl group. Aliphatic alkynes also proved to be effective for this process **91l**. The excellent availability of this multistep reaction has been well demonstrated by the generation of products **91i** and **91k** from α-azidoacetate and 2-phenylethyl azide, respectively. Under the previous mild sequential copper(I)-catalyzed azide-alkyne cycloaddition (CuAAC) and thiolization reaction conditions, the estrone derivative of an alkyne was easily transformed into the corresponding 5-thiotriazole **91m** in 52% yield.

5-Selenotriazoles **92a**–**l** were prepared using selenium as a reagent under the standard conditions in DMF or MeCN, as shown in Figure 45. The examination was extended to synthesize fused bicyclic 5-thiotriazoles **93a**–**g** [34] (Figure 46).

Copper(I) acetylide VI was formed from a terminal alkyne and CuI under basic conditions. The proposed mechanism for this work is shown in Figure 47. In the beginning, CuI reacted with sulfur to give copper sulfide I. Subsequently, the oxidative addition with a halide forms II, transformed into copper(II) thiolate IV via reductive organic group transfer and oxidation. The cycloaddition of **VI** with the azide produced the copper(I) triazolide intermediate **VI1**. Reaction of copper(II) thiolate **IV** with the copper(I) triazolide would give the intermediate **V**. Finally, reductive elimination of **V** would then lead to the expected 5-thiotriazole **89a**−**u** (Figure 47) [34].

Copper catalyzed 1,3-dipolar cycloaddition between azides and 4-allyl-2-methoxy-1-(prop-2-yn-1yloxy)benzene (**94**) in refluxing acetonitrile afforded the corresponding monocycloadduct **95a**–**g** with regioselectivity in high yields between 78 and 90% (Figure 48) [35].

5-Amino-1,2,3-triazoles **98a**–**k** were prepared from the corresponding amidines **97a**−**k** reacting with tosyl azides in a methanolic basic medium (Figure 49) [36].

Stanciu et al. [37] reported that *N*, *N′*-carbonyldiimidazole (CDI) synthesized amphiphilic esters based on dextran via a one-pot procedure based on the reaction between the polysaccharide and different substituted 1,2,3-triazoles-4-carboxylates **102a**–**f**. Firstly, the triazole derivatives **102a**–**f** were obtained through copper alkyne azide cycloaddition (CuAAC) between azide **101a**–**f** and ethyl propiolate. Basic hydrolysis of the triazole ester **102a**–**f** using KOH_(aq)_, MeOH/H_2_O gave 1,2,3-triazol-4-carboxylic acid derivatives **103a**–**f** [37]. Esterification of the dextran (polysaccharide) with the triazole ester activated in situ with 1,1′-carbonyldiimidazole (CDI) to give the dextran esters **104a**–**f** (Figure 50).

*N*-Heterocyclic carbene-copper (NHC-Cu) complexes were known as organometallic catalysts that could differentiate the reactivities of simple terminal alkynes and azides through amplified steric discrimination, allowing efficient sequential ligations of a diyne with two different azides under conditions of premixing all of the reaction partners in solution [38,39]. The interlocked NHC-CuI complexes were found as 1-TFPB and 2-TFPB (TFPB: tetrakis[3,5-bis(trifluoromethyl)phenyl] borate) (Figure 3). The rotaxane 1-TFPB catalyzed a competition reaction involving two pairs of individual alkynes and azides. Therefore, a heating mixture of the non-bulky alkyne **105**, the bulky alkyne **106**, the non-bulky azide **107**, and the bulky azide **108** in THF at 323 K for 48 h in the presence of rotaxane 1-TFPB (15 mol%), four possible triazole products were formed **109**–**112** with good selectivity. The triazole **109** was predominant, with the ratio of the triazoles **109**–**112** being 14:1:0:0 (Figure 51) [40].

Moreover, when azides **107** and **108** were reacted with diyne **113**, under the same conditions in the presence of rotaxane 1-TFPB (15 mol%) and then adding [Cu(MeCN)_4_]PF_6_ and 2,6-lutidine to the intermediate mixture bis-triazole, **114** was isolated in 74% yield (Method A) (Figure 52). When the last reaction was performed in the presence of 2-TFPB (15 mol%) under the same conditions in the dark for 48 h, the less bulky alkyne and azide had disappeared, while those of the bulky alkyne and azide remained, exhibiting good chemoselectivity toward the coupling of the less bulky parts. Irradiation of this intermediate mixture with light (350 nm, 5 min) cleaved approximately half of the macrocyclic components formed. Heating the resulting mixture (323 K, 12 h) led to the coupling of the bulky pair of alkyne/azide partners and the formation of triazole product **114** in 84% yield (Figure 52) [40].

Syntheses of 2-(4-((1-phenyl-1*H*-1,2,3-triazol-4-yl)methoxy)phenyl)isoindoline-1,3-dione derivatives **119a**–**f** were involved in three steps. 4-Aminophenol **116** reacted with phthalic anhydride (**115**) in acetic acid at 100 °C to give compound **117** (Figure 53). Propargylation of **117** afforded the corresponding 2-(4-(prop-2-yn-1-yloxy)phenyl)isoindoline-1,3-dione (**118**), which in the presence of potassium carbonate and subsequently treated with various azides via 1,3-dipolar cycloaddition (click reaction) during treatment with 10 mol% of sodium ascorbate and 10 mol% of copper sulfate. Consequently, the reaction afforded 1,2,3-triazolyisoindoline-1,3-dione derivatives **119a**–**e** in excellent 70% to 81% yield, as shown in Figure 53 [41].

Applying the same procedure mentioned, the synthesis of triazoles **123a**–**e** was also established in the same three steps. Compound **122** was obtained, which in the presence of potassium carbonate and subsequently treated with various azides and 10 mol% of sodium ascorbate and 10 mol% of copper sulfate, afforded the final compounds **123a**–**e** via 1,3-dipolar cycloaddition (Click reaction) (Figure 54) [41].

CuFe_2_O_4_@MIL-101(Cr) was used as a catalyst in synthesizing benzodiazepine triazole derivatives during the reaction of chalcones containing the acetylene group in the *o* or *p* positions **124a**–**c** with substituted azides containing both electron-withdrawing groups and electron-donating groups (Figure 55). Firstly, the chalcones **124** reacted with *o*-phenylene diamine **125** to furnish the corresponding diazepine acetylene derivatives **126**, which on click reaction underwent cyclization to give the triazole derivatives **127a**–**u** [42].

Propargylurea (**128**) underwent cyclocondensation with methyl trifluoropyruvate benzothiazolylimine (**129**) in the presence of Et_3_N to give 5-alkynyl-substituted trifluoromethyl hydantoin **130** in 67% yield. The alkyne **130** was subjected to CuAAC reaction with 2-azidoacetamides **131** to give the corresponding 1,4-substituted 1,2,3-triazoles **132a**–**f** (Figure 56) [43].

Azides reacted regioselectivity with alkynes via CuAAC 1,3-dipolar cycloaddition using CuI to form triazoles **133a**–**g** as a major product and triazoles **134a**–**g** as minor products. An exception was the reaction of methyl azide with tert-butyl prop-2-yn-1-ylcarbamate, which resulted in a mixture of triazole **133a** and 5,5′-bitriazole **135** at a ratio of 2.4:1. Under similar conditions, the longer aliphatic chain azide (3-(azidomethyl)heptane) reacted with phenylacetylene to give triazole **133h** in low yield by using copper(I) iodide as a catalyst in combination with a catalytic amount of benzoic acid furnished **133h** in a high yield (90%) (Figure 57). Bis-triazoles **137** and **139** were obtained in high yields with a faster rate of reaction via click reaction of 1,3-diazidopropane (**136**) and diazide **138** with ethyl propiolate and phenylacetylene, respectively (Figure 58). Furthermore, azide **140** (1-(2-azidoethyl)-5,6-dimethyl-1*H*-benzo[*d*][1,2,3]triazole) was reacted with diethyl acetylenedicarboxylate and methyl propiolate afforded triazole **141** (97%) and triazole **142** (94%) yields, respectively (Figure 59). Azide **143** was reacted even with a reactive dipolarophile, such as acetylenecarboxylate in *t*-BuOH in the presence of a basic cocatalyst gave triazole **144** (74%) (Figure 59) [44].

*N*-Propargylated cinnolinones **145a**–**c** reacted with benzyl azide to afford the corresponding triazole derivatives **146a**–**c** via CuAAC in CHCl_3_. It was noted that the complexes of the [(IPr)CuX] series (X = Cl, Br, I) did not exhibit catalytic activity. However, [(IMes)CuX] complexes (X = Cl, Br, I) containing a less sterically hindered ligand showed higher activity under the same reaction conditions; this was attributed to the electronic nature of the halides, and the catalytic reactivity increased in the order of Cl^−^ < Br^−^ ≈ I^−^, (Figure 60) [45].

Filimonov et al. [46] reported a one-step, eco-friendly method for synthesizing 1,2,3-thiadiazol-4-carbimidamides **149a**–**r** and 1,2,3-triazole-4-carbothioamides **150a**–**j**, during the reactions of 2-cyanothioacetamides **147a**–**g** with various types of azides **148** in water in the presence of alkali (Figure 61). Furthermore, *N*,*N*`-bis-(2-cyanothiocarbonyl)-pyrazine **147h** was reacted with sulfonyl azides **148** to give the bicyclic 1,2,3-thiadiazoles **151**–**153,** and 1,2,3-triazoles **154a**–**f** connected via a 1,1`-piperazinyl linker (Figure 62). On the other hand, 2-cyanothioacetamides **155** reacted with aromatic azides in water in the presence of alkali to afford 1-aryl-5-amino-1,2,3-triazole-4-carbothioamides **156a**–**l** (Figure 63). In contrast to aromatic azides and sulfonyl azides, 6-azidopyrimidine-2,4-diones **157a**–**c** reacted with cyanothioacetamides **147a**–**e** to give *N*-pyrimidine-6-yl-5-dialkylamino-1,2,3-thiadiazole-4-*N*-l-carbimidamides **158a**–**i**. Additionally, compounds **158a**–**i** were obtained in two step-reaction starting with 6-chloro-1,3-disubstituted-pyrimidine-2,4-dione **159** (Figure 64).

Bis(azidomethyl)-5,5-diethylpyrimidinetrione (**160**) underwent CuAAC 1,3-dipolar cycloaddition with alkyne (prop-2-yn-1-ol) to afford bis((4-(hydroxymethyl)-1*H*-1,2,3-triazol-1-yl)methyl)pyrimidinetrione (**161**) (Figure 65) [46].

The copper-mediated click reaction using 3-aminophenyl-acetylene (**162**) and benzyl azide as the starting materials gave the monotriazole **163** using a well-defined copper carbene complex [CuCl (IPr)] (IPr = 1,3-bis-(2,6-diisopropylphenyl)imidazolin-2-ylidene) as a catalyst. Compound **163** underwent diazotization and azidation followed by [3 + 2] click reaction to afford the non-symmetrical bis(triazoles) **164**. Alkylation of **164** using Meerwein’s salt (CH_3_)_3_OBF_4_ gave the dicationic pro-ligand salt **165**. The non-symmetrical triazolium salt **165** and symmetrical 1,2,3-triazolium salts [47,48,49,50] **166** and **167** were utilized to synthesize mesoionic carbene-sulfur adducts **168a**, **169b,** and **170c**. Firstly, the triazolium salts **165**/**166** were reacted with elemental sulfur in a base (KO*^t^*Bu and K_2_CO_3_) non-symmetrical mesoionic bis (NHT) compound **168** the symmetrical analog **169** in good yields, 76% and 72%, respectively. Complexes **170**–**173** were formed with the treatment of **167** with elemental sulfur in the presence of K_2_CO_3_ as a base (Figure 66) [51].

Yoshida et al. [52] reported that the double-click reaction between aryl azides and the diyne (**183**) afforded a regioisomeric mixture of bicyclo-adduct **184** (*trans*/*cis*) in an excellent yield. It was observed that 1-adamantyl azide was bulky, caused retention to the cycloaddition and gave the bis-cyclo-adduct with 18% for 4 h. The 2,6-diisopropylphenyl azide gave the bis-cyclo-adduct in quantitative yields, while the unhindered benzyl azide gave the bis-cyclo-adduct in 83% yields for 1 h. The studies were prolonged to cover the effect of the substituent in the aryl azide to clarify that the bulkiness groups enhanced the reaction rate, and the electronic nature of substituted groups in both *o*- and *p*-positions showed a limited effect on the reaction rate. In contrast, 2,6-disubstituted phenyl azides had accelerated reaction rates as the size of substituents became bulkier (the lower click ability of adamantyl azide than that of diisopropylphenyl azide was attributed to the stabilization of the azido group by hyper-conjugation with C-H bonds, which decreases the distort ability of the azido group) (Figure 67) [52]. The click ability of the alkyl and alkenyl azides with Sondheimer diyne (**174**) afforded the regioisomeric bicyclo adducts **176** (*trans*/*cis*). The studies showed that the reaction rate was faster in the case of alkyl azides than in the alkenyl azides, indicating that resonance retarded the reaction rate and that both the inductive effect and hyper-conjugation increased the reaction rate (Figure 67) [52,53].

The synthesis of triazolyl benzoxazine derivatives **179a-n** via one-pot reaction (e.g., Ugi reaction [54]) using the so-called Passerini-azide reactions (a method to prepare tetrazoles by substituting hydrazoic acid generated in situ from NaN_3_ or TMS-N_3_ (**177**), has been reported [54,55,56]. The reaction of 2-azidobenzaldehydes **176**, **177,** and isocyanides **178** gave 4*H*-3,1-benzoxazine derivatives by the in-situ formation of azide intermediate (Figure 68) [56].

The proposed mechanism for the formation of compounds **179a**–**n** was started from the Passerini-azide adduct **180,** which reacted with the palladium reagent to form the palladium–nitrogen intermediate **181** with the elimination of the nitrogen molecule. Insertion of isocyanide **178** to the formed intermediate **181** gave the three-membered ring intermediate **182**. The carbodiimide intermediate **183** was formed via reductive elimination of intermediate **182**. Finally, intermolecular cyclization of intermediate **184** resulted in the formation of the benzoxazine derivatives **179a**–**n,** as illustrated in Figure 69 [56].

Regioselective syntheses of functionalized cyclotriphosphazenes linked via 1,2,3-triazole **187a**–**e**. Firstly, 1,3,3,5,5-penta[1-(2,2-dimethyl-1,3-dioxolan-4-yl)methoxy]-1-chlorocyclotri phosphazene (**185**) reacted with 2-propyn-1-ol and 3-butyn-1-ol in the presence of NaH to give the alkynyl derivatives **186a,b**. That was followed by the cycloaddition click reaction with phenyl azide, azido acetic acid *tert*-butyl ester or diethyl (4-azidobutyl) phosphonate at 20 °C in the presence of Cu(I) (Figure 70) [57].

Another regioselective 1,3-dipolar cycloaddition reaction was established between *N*-propargyl-substituted-1,8-dioxodecahydroacridines **188a**–**d** with aromatic azides in the presence of CuSO_4_.5H_2_O/ascorbic acid as catalyst (Click reaction) in a 2:1 mixture of CH_2_Cl_2_:H_2_O at room temperature gave 1,2,3-triazole-dioxodecahydroacridine hybrids **189a**–**e** in high yields (70–86%). Furthermore, the click reaction was subjected to propargyloxy-benzaldehydes **190a**–**d** to give 1,4-disubstituted 1,2,3-triazolealdehydes **191a**–**e** in 55–88% yields (Figure 71). The application of the Hantzsch route on the corresponding 1,2,3-triazolealdehydes **191a**–**e** with 1,3-cyclohexanedione **192** produced 1,4-disubstituted 1,2,3-triazole-*O*-acridinedione **193a**–**e** (55–90% yields). Treatment of 1,2,3-triazolealdehydes **191a**–**e** with two molecules of 1,3-cyclohexanedione in triethylamine and acetic acid gave the corresponding 1,2,3-triazole-*O*-xanthenediones **194a**–**e** in 67–91% yields (Figure 71) [58].

Annulation reactions between *gem*-diamino enaminones **195** and **197** (ketene aminals) and tosyl azide furnished *N*-heterocycle fused **196a**–**q** and 5-amino side-chain **198a**–**g** functionalized 1,2,3-triazoles under transition metal-free conditions, using NaHCO_3_ as a catalyst to promote the reaction (Figure 72, Figure 73 and Figure 74). The reaction was screened to optimize the reaction conditions. Firstly, the reaction was performed using different solvents, such as water, DMSO, DMF, toluene, and MeCN; it was observed that DMSO is the solvent of choice. Additionally, the reaction was performed under different basic conditions using NaOH, *^t^*BuONa, NaHCO_3_, and DBACO; it was found that using NaHCO_3_ is favorable for obtaining high yields (Figure 72) [58].

1,2-Diacetylenic benzenes **199** were cyclized with sodium azide (NaN_3_) furnished the corresponding [1,2,3]triazolo[5,1-*a*]isoquinolines **201**, but with low regioselectivity for substrates bearing two different alkyne substituents (R^1^ ≠ R^2^) [59,60,61]. Additionally, the same triazoloisoquinolines **201** were obtained via annulation of acetylenes with (2-halo)phenyl-1,2,3-triazoles **200** under transition-metal catalyzed conditions [62,63,64,65]. On the other hand, the annulation of 2-azido-3-(2-iodophenyl)acrylates **202** to the corresponding triazolo-isoquinolines **203** was achieved using copper chloride as a transition metal catalyst in these heterocyclization [66,67] (Figure 75).

Recently, Wu et al. [68] reported that AlCl_3_ syntheses of triazoloisoquinolines via three-component Henry reaction–triazole formation–intramolecular 6-*endo*-dig cyclization were successfully achieved. Upon reacting 2-(phenylethynyl)-benzaldehyde **204a**, nitromethane, and sodium azide in the presence of Lewis acid in DMF at 100 °C, a mixture of [1,2,3]triazolo[5,1-*a*]isoquinoline **205a** and isoquinoline **206** was obtained. However, triazole **207** was obtained when an excess of AlCl_3_ was used. Among these studies, it was found that Sc(CF_3_SO_3_)_3_ and AlCl_3_ were the preferable Lewis acids to promote the formation of triazoloisoquinolones, as shown in Figure 76.

The type of solvent and Lewis acid affected the yield and the regioselectivity of the product, as in **205a**. Additionally, the substituted group affected the yield of the target product in which the presence of electron-donating groups gave higher yields than electron-withdrawing groups. On replacing the phenyl ring with pyridine ring (an electron-deficient) and naphthalene (*π*-electron delocalized group), the triazoloisoquinolones **205h** and **205i** were synthesized in moderate yields of 32 and 53%, respectively, as shown in Figure 77 [68].

Mn(OAc)_3_·2H_2_O were used as catalysts in the syntheses of bicyclic azido alcohol **208** via azide radical addition/cyclization/oxygen insertion reaction of alkyne-tethered cyclohexadienones **208** with TMSN_3_ under mild conditions. The azido alcohol **209a** was led to react with phenylacetylene via Cu-catalyzed click reaction 1,2,3-triazole **210** was obtained in 84% yield (Figure 79) [69]. The plausible mechanism for forming the azido alcohol **209a** is shown in Figure 80. It was described due to azide radical addition, then radical conjugation, and lastly, oxygen insertion process through the formation of the intermediates **A**–**E** [69].

Intramolecular azide–alkene cycloaddition of *N*-bromoalkyl indole and pyrrole derivatives **211a**–**v** resulted in the formation of polycyclic fused 1,2,3-triazoles **212a**–**v** [70]. As a model example, the reaction progress was investigated to determine the optimized conditions via the reaction of **211a** (0.5 mmol) with sodium azide (0.6 mmol) in ethanol at room temperature for 20 h and under catalyst-free conditions. The reaction proceeded smoothly to give (6,7-dihydro-5H-[1,2,3]triazolo[5`,1`:3,4][1,4]diazepino[1,2-*a*]indol-1-yl)(phenyl)-methanone **212a** in 64% yield (Figure 81). When the reaction was first applied to the seven-membered ring annulated indole by varying substituents (R^1^) on the benzoyl group. It was observed that electron-donating groups, such as amine, methoxy, hydroxyl, and isobutyl, under the optimal reaction conditions, gave the desired products in 65–91% yields (Figure 82). Similarly, halogen substituents were also produced the appropriate products (**212b**–**d**, **212i**, and **212m**) in good to high yields (73–81%). However, highly electron-poor substituents, such as CF_3_, showed lower efficiency (**212k**, 69% yield), and the reaction of 3,5-ditrifluoromethyl acetophenone failed to give the corresponding product. Additionally, the investigation was extended to prepare six-membered ring annulated indoles by using *N*-bromoethyl substituent on the indole under the optimized conditions. It was shown that electron-donating and electron-withdrawing groups produced the corresponding fused polycyclic *N*-heterocycles (**212o**–**t**) in slightly lower yields (72–81%). Alkyl groups on the ketone derivatives led to the desired products **212u** and **212v** in good yields (77% and 72%, respectively) [70].

Ugi four-component reaction/alkyne–azide cycloaddition reaction was applied to synthesize triazoloquinoxalines. Reacting 2-azidobenzenamines **213**, isocyanide **178**, aldehydes, and propiolic acids **214** afforded [1,2,3]triazolo[1,5-*a*]quinoxalin-4(5*H*)-ones **216a**–**s** via the formation of Ugi adducts **215**. The cyclization occurs via an alkyne–azide cycloaddition reaction (Figure 83) [71].

Gangaprasad et al. [72] reported the syntheses of 1,2,3-triazole fused benzooxazepine and benzodiazepine analogs **218a**–**q** via one-pot azide substitution and intramolecular azide-olefin **217** oxidative cycloaddition sequence under metal-free conditions (Figure 84) [72].

Aly et al. [73] reported that copper(I)-catalyzed azide-alkyne [3+2] dipolar cycloaddition reaction (CuAAC) between **219a**–**d** and **220** to afford the target hybrids **221a**–**d**, in good to excellent yields depending on the concentration of catalyst (Figure 85). Additionally, the target compounds **221a**–**d** were synthesized, in very good yields, via the reaction of 4-{[1-(2-oxo-1,2-dihydroquinolin-4-yl)-1*H*-1,2,3-triazol-4-yl]methoxy}benzaldehydes **222a**–**d** [73] with acetophenone (Figure 85).

Similarly, doubly derivatized chalcones were prepared by the interaction between (*E*)-1,3-bis[4-(prop-2-yn-1-yloxy)phenyl]prop-2-en-1-one (**223**) and 4-azidoquinolin-2(1*H*)-ones **219a**–**d** in the presence of CuAAC to obtain 1,2,3-triazoles **224a**–**d** [73]. The 1,2,3-triazoles **224a**–**d** were also synthesized by the reactions of aldehydes **225a**–**d** with 4-{4-[(4-acetylphenoxy)-methyl]-1*H*-1,2,3-triazol-1-yl}-quinolin-2(1*H*)-ones **226a**–**d** in basic medium, as shown in Figure 86.

Aly et al. [74] also reported that the synthesis of hybrids **228a**–**g** through click chemistry which is a powerful tool for a quick, highly selective, and reliable access to a reaction product with high yields. The [3+2] cycloadditions of 4-azidoquinolin-2(1*H*)-ones **219a**–**d** with 4-(prop-2-yn-1-yloxy)quinolin-2(1*H*)-ones **227a**–**c**, gave the corresponding 4-((1-(2-oxo-1,2-dihydroquinolin-4-yl)-1*H*-1,2,3-triazol-4-yl)methoxy)quinolin-2(1*H*)-ones **228a**–**g** (Figure 87). Compounds **228a**–**c** were found to be the most active antiapoptotic hybrids with significant measurements for the antioxidant parameters (malondialdehyde (MDA), total antioxidant capacity (TAC), and the apoptotic biomarkers (testicular testosterone, tumor necrosis factor (TNFα) and caspase-3) in comparison to the reference. A preliminary mechanistic study was performed in order to improve the antiapoptotic activity through caspase-3 inhibition. A compound assigned as 6-methoxy-4-(4-(((2-oxo-1,2-dihydroquinolin-4-yl)oxy)methyl)-1*H*-1,2,3-triazol-1-yl)quinolin-2(1*H*)-one (**228c**) was selected as a representative of the most active hybrids in comparison to *N*-acetyl cysteine (NAC). Assay of cytochrome *C* for **228c** revealed a down expression of cytochrome *C* level by about 3.54 fold, comparable to NAC (4.13 fold). In caspases-3,8,9 assays, **228c** was found to exhibit more potency and selectivity toward caspase-3 than other caspases. Testicular histopathological investigation was carried out on all targeted compounds **228a**–**g,** indicating a significant improvement in spermatogenesis process for compounds **228a**–**c** if compare with the reference relative to the control [74].

### 4.6. Synthesis of Tetrazole Ring

One-pot syntheses of 5-substituted 1*H*-tetrazole derivatives **229a**–**j** [75] were achieved using a dimethyl sulfoxide–nitric acid combination in an aldehyde, hydroxylamine combination hydrochloride, and sodium azide under mild conditions (Figure 88). The proposed mechanism is illustrated in Figure 89 [75].

Heterocyclization of 1,2,4-triazol-3-amine **230** and 3-amino-1-*tert*-butyl-1,2,4-triazole **231** was established via alkylation of 3-amino-1,2,4-triazole **230** using *t*-BuOH-HClO_4_ with triethyl orthoformate and sodium azide in absolute ethanol. The reaction gave 1-(1,2,4-triazol-3-yl)-1*H*-tetrazole **232** and 1-(1-*tert*-butyl-1,2,4-triazol-3-yl)-1*H*-tetrazole **233**, respectively, as depicted in Figure 90 [76].

Grinding a mixture of Schiff bases: 4-(3-hydroxybenzylideneamino)antipyrine and 4-(4-nitrobenzylideneamino)antipyrine **234a,b** with sodium azide (NaN_3_) gave the corresponding tetrazoles **235a,b** (Figure 91) [77].

Cobalt nano-particles, a heterogeneous catalyst, catalyzed the synthesis of tetrazoles **236a**–**j** from a multicomponent reaction of amines, sodium azide, and triethyl orthoformate under solvent-free conditions at 100 °C (Figure 92). The reaction was screened for the effects of the amount of both catalyst and solvent; it was found that carrying the reaction using 50 mg of the catalyst under solvent-free conditions gave the tetrazole **236a** a 96% yield. The proposed mechanism is illustrated in Figure 93, which involved condensation between the amine and ethyl orthoformate followed by cycloaddition ([1,3]-dipolar cycloaddition) of azide and imine to give the tetrazole product [78].

Reaction of the chloropyrimidine (7-chloro-3-methyl-1-phenyl-1*H*-pyrazolo [4’,3’:4,5] thieno[3,2-*d*]pyrimidine) (**237**) with sodium azide in DMF in presence of NH_4_Cl gave the tetrazole derivative **238**, which was identified as (7-methyl-9-phenyl-9*H*-pyrazolo-[4’,3’:4,5]thieno[2,3-*e*]tetrazolo[1,5-*c*]pyrimidine) (Figure 94) [79].

Ag/Fe_3_O_4_ nanocomposite catalyzed the synthesis of 5-(3-bromophenyl)amino-1*H*-tetrazole**240** from 3-bromophenyl cyanamide **239** and sodium azide in DMF at 110 °C [78]. Aminotetrazole–palladium (II) complex **243** was prepared via the nucleophilic substitution between 5-(3-bromophenyl)amino-1*H*-tetrazole **240** and Fe_3_O_4_@SiO_2_@(CH_2_)_3_-Cl (**241**), followed by incorporation of the Pd-ions using PdCl_2_.2H_2_O in EtOH under reflux for 24 h [80] (Figure 95).

Trose et al. [79] have reported that the reaction of *N*-Heterocyclic carbene (NHC)-based copper azide complex [Cu(N_3_)(IPr)] (**244**) (IPr = *N*,*N*`-bis[(2,6-(di-isopropyl)phenyl)]-imidazole-2-ylidene) with dimethyl acetylenedicarboxylate (as an activated alkyne), produced triazolate copper complex **245** (Figure 96). However, complex **244** was found that it was reacted with the activated *p*-toluenesulfonyl cyanide (**246**) to give the tetrazole complex **247** in quantitative yield upon mixing (98%). In contrast, the reaction of complex **244** with the less activated 4-(trifluoromethyl)benzonitrile (**248**) needed heating and longer reaction times (50 °C, 16 h) to form the bis tetrazole complex **249** in high yield (93%) (Figure 96) [81].

### 4.7. Synthesis of Thiadiazole

The more reactive thioamide **250** was reacted with tosyl azide in the presence of Et_3_N at room temperature to afford 1,2,3-thiadiazole **251** (30 and 22% yield) together with compounds **252**. At the same time, the thioamide **147c** reacted with tosyl azide (R = *p*-Me-C_6_H_4_) to produce thiadiazole **253** (100 and 90% yield) (Figure 97) [36].

### 4.8. Synthesis of Pyridine and Isoquinoline Derivatives

Singam et al. [82] reported the regioselective arylnicalation of *ortho* functional diaryl acetylene **254a** with Ar(BOH)_2_ **255a**–**p** to synthesize substituted di-aryl isoquinolines **256a**–**p** (Figure 98).

Additionally, *ortho* diarylacetylene derivatives **254b**–**m** were investigated under the same reaction conditions, which on reacting with Ph(BOH)_2_ **255a** gave the desired product **256q**–**b`** in high yields, as depicted in Figure 99.

The 5,6-diarylnicotinates **258a**–**e** were performed from enynyl azides **257** with **255a,d,g,h,q** under the same standard conditions (Figure 100) [82].

Pd-PEPPSI-IPr was used as a catalyst to reach the optimal reaction conditions during the reaction of acetophenone-*N*-acetylhydrazone (**259a**) and (1-azidovinyl) benzene (**260a**) (Figure 101) [83]. It was concluded that toluene was the best solvent of choice, and heating to 100 °C gave 81% yield of **261a** (Figure 101)

The procedure showed that *N*-acetyl hydrazones **259** were screened to react with (1-azidovinyl)benzene (**260a**), as shown in Figure 102. Variation from alkyl aryl ketones to benzophenone and cycloalkyl aryl ketones hydrazones reacted smoothly with **260a** to afford isoquinolines **261a**–**ab** in 60–81% yields (Figure 102) [83]. The C-H functionalization occurred regioselectively at the less hindered site for *meta*-substituted substrate (Me, **259k**), yielding a mixture of two isomers, **261k** (major) and **261l** [83]. Either electron-donating (Me, Bu, Ph, OMe, OPh) or electron-withdrawing (F, Br, Cl, CN) group on the *para*-position of the phenyl ring of acetophenone *N*-acetylhydrazones were transformed to the desired products in moderate yields.

The reaction of various vinyl azides **260a**–**p** with *N*-acetyl hydrazone **259a** under the standard reaction conditions was examined (Figure 103). Fused isoquinolines **261ac**–**ap** were obtained via the same previous procedure [83].

The transformations of 1-tetralone, 1-benzosuberone hydrazones **262a**–**d** proceeded smoothly to give the desired polycyclic product **263a,b** in moderate yields. Moreover, chroman-4-one and thiochroman-4-one-hydrazone substrates were converted to polyheterocyclic products **263c,d** in 88% and 91% yields (Figure 104).

7-Methoxyflavanone **264** was reacted with acetohydrazide **265** to hydrazone **266**, which, when treated with vinyl azide **260a**, provided the isoquinoline product **267** in 51% yield (Figure 105) [83].

### 4.9. Synthesis of Phenanthridine

Phenanthridines **269a,b** were synthesized using the catalytic system of FeCl_2_/*N*-heterocyclic carbene (NHC) SIPr-HCl (1,3-bis-(2,6-diisopropylphenyl)imidazolinium chloride) from 9-azidofluorenes **268,b** via 1,2-aryl migration (Figure 106) [84].

### 4.10. Synthesis of Imidazoindoles

Jin and others [85] reported that the multicomponent reaction of sulfonyl azides, alkynes **270**, and allylamines **271** was catalyzed by copper iodide in the presence of triethylamine in DMSO/K_2_CO_3_ and dimethyl ethylenediamine as a ligand (**L**), affording 2,3-dihydro-1*H*-imidazo[1,2-*a*]indoles **272a**–**t** (Figure 107). Four C–N bonds were formed by way of azide-alkyne cycloaddition (CuAAC) and double Ullmann-type coupling reactions in a one-pot process, as illustrated in the reaction mechanism (Figure 107) [85].

The proposed mechanistic steps were proposed, as shown in Figure 108. First, a copper-catalyzed azide-alkyne cycloaddition reaction (CuAAC) takes place to generate intermediate **A**, which then transforms to ketenimine **B** by the extrusion of N_2_. Nucleophilic addition of 2-bromoprop-2-en-1-amine (**270a**) to intermediate **B** affords carboxamidine **C** and/or its tautomer. Finally, a consecutive coppercatalyzed C–N coupling reactions proceeds to provide 2,3-dihydro-1*H*-imidazo[1,2-*a*]indole **272a** (Figure 108) [85].

### 4.11. Synthesis of Quinazoline Derivatives

Kumar et al. [86] reported the tandem synthesis of 2-quinazoline carboxylates **275a**–**m** using 2-(azidomethyl)phenyl isocyanides **273a**–**m** along with carbazates **274** in the presence of Mn(OAc)_3_·2H_2_O (Figure 109).

The postulated mechanism is illustrated in Figure 110. Initially, the Mn(OAc)_3_·2H_2_O-assisted homolysis of *tert*-butyl hydroperoxide (TBHP) generates the *tert*-butoxy and the *tert*-butyl peroxy radicals. Bond cleavage of the C-N in methyl carbazate **274** forms the alkoxycarbonyl radical (**B**), which loses molecular N_2_. Radical (**B**) then attacks the R–NC bond of 2-(azidomethyl)phenyl isocyanide (**273a**) to form an imidoyl radical intermediate (**C**). The intermediate **C** then undergoes intermolecular cyclization with the azido group to give a cyclized aminyl radical (**D**) by nitrogen loss. Finally, a hydrogen abstraction of the radical intermediate (**D**) leads to the desired product (**275a**) (Figure 110) [86].

### 4.12. Synthesis of Borane Containing Heterocycles

#### 4.12.1. Synthesis of Diborylaniline and Diboryl-Fused Pyrimidine

Prieschl et al. [87] reported that the c-nitrogen insertion of aryl azides into the B–B bond of electron-rich cyclic l-hydridodiboranes **276** was stabilized by one *N*-heterocyclic carbene (NHC) ligand leads to the expansion of the central C_3_B_2_ ring, yielding unsymmetrical polyheterocyclic 1,1-diboryltriazenes **278** and **279** via the intermediate **277**. The 2-benzyl-bridged analogs undergo further NHC ring expansion and thermally-induced loss of N_2_ to give polyheterocyclic diborylanilines **280** (Figure 111) [87].

#### 4.12.2. Synthesis of Triazaphosphaborolidine

1-(Benzo[*d*][1,3,2]dioxaborol-2-yl)-2-(diphenylphosphino)-1,2-diphenylhydrazine (280) showed frustrated Lewis pairs (FLPs) reacted with benzyl azide, forming 4’-benzyl-1’,2’,3’,3’-tetraphenylspiro[benzo[*d*][1,3,2]dioxaborole-2,5’-[1,2,4,3,5]triazaphosphaborolidin]-3’-ium-12-uide (**281**) in 73% yield (Figure 112) [88].

### 4.13. Synthesis of Aluminum-Containing Heterocyclic

Drescher and others [89] reported the ring expansion of alumina cyclopentadienes (alumoles) on treatment with organic azides. Treatment of alumole **282** and trimethylsilyl azide in benzene at 60 °C gave cycloadduct **283** in 61% yield, while mesityl (Mes = 2,4,6-Me_3_C_6_H_2_) or 2,6-diphenylphenyl azide forming the aza-cycloadducts **284** (31%) or **285** (73%), respectively (Figure 113) [89].

### 4.14. Synthesis of Phosphorus-Containing Heterocycles

#### 4.14.1. Synthesis of Triazphosphocine

Treatment of 1-(di-tert-butylphosphino)-3-methyl-1,2,3,4-tetrahydroquin azoline (**286**) with phenyl azide gave benzo[*g*][1,3,5,2]triazaphosphocine (**287**) instead of phosphazide derivative ((*E*)-1-(di-tert-butyl(phenyltriaz-2-en-1-ylidene)phosphoranyl)-3-methyl-1,2,3,4-tetrahydroquinazoline) (**288**). The reaction of compound **286** with phenyl azide takes place on P(III) of compound **286** was believed to give P(V) product phosphazide **288**, which underwent ring enlargement to give benzotriazaphosphocine **287**. Derivative **289** reacted readily with phenyl azide to give compound **290** in a high yield (Figure 114) [90].

#### 4.14.2. Synthesis of Benzothiazaphosphole

The reaction of *ortho*-phosphinoarenesulfonyl fluorides **291** with trimethylsilyl azide resulted in benzo-1,2,3-thiazaphosphole **292** [91]. To optimize the reaction condition, it was found that a mixture of acetonitrile and 10 equivalents of trimethylsilyl azide at 60 °C was the optimal reaction chosen condition (Figure 115) [92]. The three possible mechanistic pathways (A), (B) and (C) for forming the benzo-thiazaphosphole **292a** are illustrated in Figure 116 [91].

## 5. Conclusions

In summary, the azido group in organic substrates are effectively served in the synthesis of various heterocycles through different mechanistic steps, such as one-pot reactions, nucleophilic additions (such as Aza-Michael addition), cycloaddition reactions (such as [3+2] cycloaddition), mixed addition/cyclization/oxygen, and insertion reactions of C-H amination. The selectivity of the chosen catalyst plays an important role in the chemoselectivity favoring C−H and C-N bonds, as it can be seen that organic azides have been used in the synthesis of various types of natural products producing good to excellent yields. Most indicative is the utility of organic azides in the synthetic procedures of fused heterocycles, such as quinazoline derivatives along with organo-metal heterocycles (i.e., phosphorus-, boron-, and aluminum-containing heterocycles). This review focused on synthesizing various heterocycles using azide chemistry and mechanistic aspects.

## Data Availability

Data is contained within the article.

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
