# Peer review of "Azides in the Synthesis of Various Heterocycles"

_molecules, 2022, doi:10.3390/molecules27123716_

Round 1
Reviewer 1 Report
This review article depicts a literature survey on Azides in the synthesis of various heterocycles in a concise way. Report focused on synthesizing various heterocycles from organic azides and mechanistic aspects. Also they have nicely included the synthetic procedures of fused heterocycles such as quinazoline derivatives and organometal heterocycles. The authors nicely covered the reported literatures in the last decade on this topic. Apart from contents and literature throughout the manuscript there are many random mistakes. It requires recheck all the scheme including substrate structure, yield of that substrate, catalytic cycle. The colour of compounds is totally un-uniformed and last please align all the structure for better understanding and attracting. After addressing all issues that article can be considered for publication in Molecules.
Recheck all the scheme including substrate structure, yield of that substrate, catalytic cycle few of them I mentioned
- L3-L9 in place of L3-L8
- Scheme 8. Please include the ‘n’ belongs to? And make the uniformity in the substrate please align all the structure
- Redraw the catalytic cycle in Scheme 10
- Mention the ‘dr’ in each substrate in Scheme 11, 12, 13.
- Scheme 16 Substrate e is not in the reported publication
- Scheme 17 Substrate I, j, k is not in the reported publication
- Scheme 18 Substrate n, q, x is not in the reported publication
- Scheme 19 Clean up all the structure redraw the cycle and please count the no. of carbon in each molecule and correct it.
- Scheme 21 recheck and correct it.
- Scheme 22 Incorrect structure (37l, 37m) incorrect yield (37u, 37z).
- Scheme 25 Incorrect structure e, f, g.
- Scheme 34 recheck the yields of substrate f
- Scheme 35 recheck substrate q, v.
- Scheme 38 redraw the transition state correctly.
- Scheme 45recheck substrate m.
- Scheme 47 recheck substrate f.
- Scheme 48redraw the catalytic cycle.
Author Response
This review article depicts a literature survey on Azides in the synthesis of various heterocycles in a concise way. Report focused on synthesizing various heterocycles from organic azides and mechanistic aspects. Also they have nicely included the synthetic procedures of fused heterocycles such as quinazoline derivatives and organometal heterocycles. The authors nicely covered the reported literatures in the last decade on this topic.
Many Thanks
Apart from contents and literature throughout the manuscript there are many random mistakes. It requires recheck all the scheme including substrate structure, yield of that substrate, catalytic cycle.
Answer: Was done
The colour of compounds is totally un-uniformed and last please align all the structure for better understanding and attracting. After addressing all issues that article can be considered for publication in Molecules.
Answer: The colors of schemes were changed to accommodate the reactions. For example, azide compounds take blue colors, whereas products take red color
Recheck all the scheme including substrate structure, yield of that substrate, catalytic cycle few of them I mentioned
I noted a mistake in numbering the schemes starting from Scheme 4. So the Scheme numbering was decreased by one. The same mistake was noted at the end of the review, and the references exceeded by one. So that the number of references became the same. Everything becomes OK
- L3-L9 in place of L3-L8
Was done
- Scheme 8. Please include the ‘n’ belongs to? And make the uniformity in the substrate please align all the structure
(Becomes Scheme 7) and was corrected and done.
- Redraw the catalytic cycle in Scheme 10 (becomes Scheme 9)
Was done
- Mention the ‘dr’ in each substrate in Schemes 11, 12, 13 (Becomes 10, 11, 12)
Was done
- Scheme 16 (becomes Scheme 15) Substrate e is not in the reported publication
Was corrected
- Scheme 17 (becomes Scheme 16) Substrate I, j, k is not in the reported publication
Was corrected
- Scheme 18 (becomes Scheme 17) Substrate n, q, x is not in the reported publication
Answer: The structures were corrected
- Scheme 19 (becomes Scheme 18) Clean up all the structure redraw the cycle and please count the no. of carbon in each molecule and correct it.
Was corrected
- Scheme 21 (becomes Scheme 20) recheck and correct it.
Was corrected
- Scheme 22 (becomes Scheme 21) Incorrect structure (37l, 37m) incorrect yield (37u, 37z).
R was put in state of X. The yields of 37u and 37z were corrected.
- Scheme 25 (becomes Scheme 24) Incorrect structure e, f, g.
Were corrected
- Scheme 34 (becomes Scheme 33) recheck the yields of substrate f
Was corrected
- Scheme 35 (Scheme 34) recheck substrate q, v.
Were corrected
- Scheme 38 (Scheme 37) redraw the transition state correctly.
Was corrected
- Scheme 45 (Scheme 44) recheck substrate m.
Was corrected
- Scheme 47 (becomes 46) recheck substrate f.
Was corrected
- Scheme 48 (becomes 47) redraw the catalytic cycle.
Was done
Reviewer 2 Report
The observations are very numerous and refer to many aspects of the manuscript.
The manuscript contains a lot of data that is reproduced without the authors processing and synthesizing them, based on their purpose.
A series of paragraphs superficially present data from the literature, data which, to be understood, need access to the bibliographic source. Many acronyms are not explained.
Many of the data presented in the text are not correctly presented in the reaction diagrams. It refers in the text to compounds with a certain numerical code that are not found in the reaction schemes.
Not all reaction schemes are presented in the manuscript, or several confusions are made.
The same substituents with different notations appear in the same reaction scheme, e.g. Ar or Ph.
Substitutes in many formulas are not explained.
All compounds, with their structural formulas, are presented on the reaction schemes, in full pages, a presentation that has no relevance for the work, because is the general formula is enough.
The chemical formulas in the reaction schemes that show the probable reaction mechanisms should also include the pairs of non-participating electrons, to make it easier for the reader to follow and understand the chemical transformation.
The manuscript is very difficult to read, with many useless things that do not serve the purpose proposed by the authors and that make it uninteresting to specialists.
The authors should take into consideration that the manuscript should be addressed to specialists in the field of organic chemistry, not strictly to specialists in whose works the authors have tried to make a synthesis. I also suggest the need for a comparative analysis of the data presented in the manuscript, as far as possible: the same type of reactions, under different catalysis conditions, electronic and steric factors that influence reactivity, which determine the regioselectivity and stereoselectivity of reactions etc.
I consider that the paper requires a major revision and a rigorous and correct presentation of the information, the elimination of confusions, the correction of mistakes both in form and in substance.
Author Response
- I don't feel qualified to judge about the English language and style
- Answer: The English of the paper was polished
- Comments and Suggestions for Authors
- The observations are very numerous and refer to many aspects of the manuscript.
- The manuscript contains a lot of data that is reproduced without the authors processing
and synthesizing them, based on their purpose.
- Answer: I have many papers of triazoles and I added two of them (yellow highlights)
- A series of paragraphs superficially present data from the literature, data which, to be understood, need access to the bibliographic source. Many acronyms are not explained.
- Many of the data presented in the text are not correctly presented in the reaction diagrams. It refers in the text to compounds with a certain numerical code that are not found in the reaction schemes.
- Not all reaction schemes are presented in the manuscript, or several confusions are made.
Thanks. We altered this when it was appropriate
- The same substituents with different notations appear in the same reaction scheme, e.g. Ar or Ph.
Thanks. We altered this when it was appropriate.
- Substitutes in many formulas are not explained.
Thanks. We altered this when it was appropriate.
- All compounds, with their structural formulas, are presented on the reaction schemes, in full pages, a presentation that has no relevance for the work, because is the general formula is enough.
- We altered this when it was appropriate.
- The chemical formulas in the reaction schemes that show the probable reaction mechanisms should also include the pairs of non-participating electrons, to make it easier for the reader to follow and understand the chemical transformation.
- We used sometime electron-pushing arrows and we included them.
- The manuscript is very difficult to read, with many useless things that do not serve the purpose proposed by the authors and that make it uninteresting to specialists.
- The authors should take into consideration that the manuscript should be addressed to specialists in the field of organic chemistry, not strictly to specialists in whose works the authors have tried to make a synthesis. I also suggest the need for a comparative analysis of the data presented in the manuscript, as far as possible: the same type of reactions, under different catalysis conditions, electronic and steric factors that influence reactivity, which determine the regioselectivity and stereoselectivity of reactions etc.
- In order to keep the number of pages, we would propose the constant as is stands.
- I consider that the paper requires a major revision and a rigorous and correct presentation of the information, the elimination of confusions, the correction of mistakes both in form and in substance.
Answer: We did that
Reviewer 3 Report
In manuscript molecules-1737258 the corresponding authors A. A. Aly, S. Braese and their coworkers present a review article on the use of azides in the preparation of organic compounds containing heterocyclic rings. The authors review 90 references on 84-pages containing 114 schemes and 2 figures. The review is written concise; the schemes are drawn very clear. The colors in the schemes are used were necessary. Examples and mechanisms are included where appropriate. This is a very good manuscript, I support publishing it in the MDPI journal Molecules, after the following points are considered, in particular major points 1a and 1b:
1a. The authors state (last sentence of the abstract) that the review focuses on new literature. Please explain whether the review is comprehensive, or if not, how did you choose to include some references and other not.
1b. Chapter 3 is divided in more than 10 smaller parts based on the type of target heterocycle. I propose to consider including a small overview of the target heterocycles at the beginning of that chapter. Maybe a short text and a scheme highlighting the structures of all the target heterocycles. I think that this would add to the quality of the presented material.
- The authors claim that organic azide has five mesomeric structure, but Figure 1 shows only four mesomeric structures, Ia-Id?
- The Journal of the Chemical Society, Perkin Transactions 1 was discontinued quite some time ago. Ref 10 was published in 2002 and not in 2022.
4. Chapter one. The title of the chapter implies that only some synthetic procedures (for the preparation of organic azides) are included, while others are neither included nor even mentioned. Please explain the choice of the included procedures. In particular, it seems intuitive to include chapters 1.2 (broad scope) or 1.3 (high yield). The reason for the inclusion of chapters 1.1 and 1.4 remains unclear.
Author Response
I appreciate the reviewer's comments
Comments and Suggestions for Authors
In manuscript molecules-1737258 the corresponding authors A. A. Aly, S. Braese and their coworkers present a review article on the use of azides in the preparation of organic compounds containing heterocyclic rings. The authors review 90 references on 84-pages containing 114 schemes and 2 figures. The review is written concise; the schemes are drawn very clear. The colors in the schemes are used were necessary. Examples and mechanisms are included where appropriate. This is a very good manuscript, I support publishing it in the MDPI journal Molecules, after the following points are considered, in particular major points 1a and 1b:
Answer: Thanks
1a. The authors state (last sentence of the abstract) that the review focuses on new literature. Please explain whether the review is comprehensive, or if not, how did you choose to include some references and other not.
Answer: Was declared that in yellow highlights. The ward comprehensive was delated. As this review deals with some interesting chosen examples……..
1b. Chapter 3 is divided in more than 10 smaller parts based on the type of target heterocycle. I propose to consider including a small overview of the target heterocycles at the beginning of that chapter. Maybe a short text and a scheme highlighting the structures of all the target heterocycles. I think that this would add to the quality of the presented material.
Was shown in the paragraph under the title Organic azides in the synthesis of heterocycles and the new Figure 2 (yellow highlights)
- The authors claim that organic azide has five mesomeric structure, but Figure 1 shows only four mesomeric structures, Ia-Id?
Was corrected
- The Journal of the Chemical Society, Perkin Transactions 1 was discontinued quite some time ago.
That was in Ref 9. The year is 2002, where was Perkin Trans was found (It was corrected in yellow highlight)
- Ref 10 was published in 2002 and not in 2022.
Ref 10 is in 2003 and it is correct
- Chapter one. The title of the chapter implies that only some synthetic procedures (for the preparation of organic azides) are included, while others are neither included nor even mentioned. Please explain the choice of the included procedures.
Various methods have dealt with the preparation of azides. We selective choosing some of them ….that all.
- In particular, it seems intuitive to include chapters 1.2 (broad scope) or 1.3 (high yield). The reason for the inclusion of chapters 1.1 and 1.4 remains unclear.
The reviewer' comment is correct, however as I mentioned these are some chosen examples. Interestingly, you have to choose methods of poor synthetic procedure and you have to mention how the synthetic procedure has been developed to give good or excellent yields.
Round 2
Reviewer 2 Report
Scheme 7: which of the compounds 11a-k is tryptamine? In scheme 7 there are also pyroindole derivatives. Scheme 8: the substituents appear on the azide aromatic ring, in the ortho/ meta/para positions, not only in the para position, as mentioned in the text. Scheme 9 shows the formation of furoindoles, not pyrroindoles, as the authors mention. The acronym NMO (line 185) and the acronym DBU (line 192) must be explained. What happens with the R3 rest in Figure 15 and Figure 17? Scheme 18: The bond between N and Ni is a double bond? What about the valence of Ni? The acronym CSA (line 241) and the acronym DABCO (line 242) must be explained. Schemes 20 and 21 can be put together, in my opinion. Lines 351 and 360: cynnamyl azide instead of cinnamoyl azide. The authors are asked to explain why the reaction in Scheme 38 is regiospecific, and that in Scheme 40 is regioselective. The acronym CuAAC (line 457) needs to be explained. The structure of compound 113 needs to be checked. I think an oxygen atom is missing, given the structure of compound 114. Scheme 55: The authors must check the position of the C=N bond in the diazepine ring. Regardless of the sequence of the two nucleophilic attacks of o-phenylenediamine, the double bond is formed to the other nitrogen atom. Scheme 59: To obtain compound 142, R2 should be COOCH3, instead of CH3. Scheme 66: Compounds 164, 165 and 168 contain the remaining benzyl instead of butyl. Figure 67: The abbreviation Ar for adamantyl and benzyl does not seem appropriate to me. You can put an R. Compound h in the same scheme has Ar= o-ipropyl (I think it's a mistake). Schemes 67 and 68 can't be reunited? Scheme 72: What does ortho/meta/para(OCH2CCH) mean? Schemes 73 and 74 cannot be reunited? Schemes 82 and 83 cannot be reunited? The acronyms MDA, TAC (line 871), TNFalfa (line 872), NAQC9line 878) must be explained. Schemes 89 and 90: the meanings of the residues Ar and R must be explained. Abbreviations must be explained at first appearance in the text, for the benefit of the reader. In conclusion, the material must be read carefully by the authors, and where appropriate, it is necessary to verify the information from the bibliographic source.
Author Response
Herein are our corrections, hoping to get your acceptance ASAP.
- Scheme 7: which of the compounds 11a-k is tryptamine? In scheme 7 there are also pyroindole derivatives.
Answer: Done, the types of derivatives 11 have been specified; tryptophols (11a-n) and tryptamines (11o-r). An appropriated phrase was inserted in the text. Also, the title of Scheme 7 was changed to comprise both pyroindole and furoindoline derivatives.
- Scheme 8: the substituents appear on the azide aromatic ring, in the ortho/ meta/para positions, not only in the para position, as mentioned in the text.
Done; Scheme 8 was modified and corrected.
- Scheme 9 shows the formation of furoindoles, not pyrroindoles, as the authors mention.
Answer: Done; the word "pyrroloindolines" was omitted (red highlights).
- The acronym NMO (line 185) and the acronym DBU (line 192) must be explained.
Done; these abbreviations have been clarified (red highlight)
- What happens with the R3 rest in Figure 15 and Figure 17?
Thanks for your interesting comment. The R3 motif is the part that has undergone cyclization; forming the pyrrolidine ring. Therefore, the color of this motif has been changed to black to be more obvious.
- Scheme 18: The bond between N and Ni is a double bond? What about the valence of Ni?
Done; the double bond was replaced by a single bond.
- The acronym CSA (line 241) and the acronym DABCO (line 242) must be explained.
Done; these abbreviations have been clarified.
- Schemes 20 and 21 can be put together, in my opinion.
Thanks for your suggestions, Scheme 20 described the trials for achieving the optimal conditions for the assembly of the targeted compound (37a), while Scheme 21 comprised the targeted derivatives along with their yields. Thus, in my opinion, keeping them separate might be better than combining them.
- Lines 351 and 360: cynnamyl azide instead of cinnamoyl azide.
Answer: Rereferring to the ref, it is cinnamyl azide
- The authors are asked to explain why the reaction in Scheme 38 is regiospecific, and that in Scheme 40 is regioselective.
Answer: It was declared the difference between two schemes (red highlights).
- The acronym CuAAC (line 457) needs to be explained.
Done; Done; this abbreviation has been clarified (red highlights).
- The structure of compound 113 needs to be checked. I think an oxygen atom is missing, given the structure of compound 114.
Thanks for your great observation, the structure of compound 13 has been corrected.
- Scheme 55: The authors must check the position of the C=N bond in the diazepine ring. Regardless of the sequence of the two nucleophilic attacks of o-phenylenediamine, the double bond is formed to the other nitrogen atom.
The position of the C=N bond in the diazepine ring was checked and found to be corrected according to the reported reference: DOI: 10.1002/aoc.5782; [42].
- Scheme 59: To obtain compound 142, R2 should be COOCH3, instead of CH3.
You're absolutely right; this substituent was corrected.
- Scheme 66: Compounds 164, 165 and 168 contain the remaining benzyl instead of butyl.
Answer: That is OK as drawn
- Figure 67: The abbreviation Ar for adamantyl and benzyl does not seem appropriate to me. You can put an R. Compound h in the same scheme has Ar= o-ipropyl (I think it's a mistake).
Done; the abbreviation "Ar" was replaced by "R". Also, the "o-ipropyl" was corrected to be R = o-iPr-C6H4.
- Schemes 67 and 68 can't be reunited?
Answer: was added and the numbering of scheme was reduced by one.
- Scheme 72: What does ortho/meta/para(OCH2CCH) mean?
Thanks; The Scheme was corrected (i.e. the substituents).
- Schemes 73 and 74 cannot be reunited?
Thanks for your suggestions, Scheme 73 described the trials for achieving the optimal conditions for the assembly of the targeted compound (196a), while Scheme 74 comprised the targeted derivatives along with their yields. Thus, in my opinion, keeping them separate might be better than combining them.
- Schemes 82 and 83 cannot be reunited?
Thanks for your suggestions, Scheme 82 described the trials for achieving the optimal conditions for the assembly of the targeted compound (134a), while Scheme 83 comprised the targeted derivatives along with their yields. Thus, in my opinion, keeping them separate might be better than combining them.
- The acronyms MDA, TAC (line 871), TNFalfa (line 872), NAQC9line 878) must be explained.
Done; these abbreviations have been clarified (red highlights).
- Schemes 89 and 90: the meanings of the residues Ar and R must be explained.
For Schemes 89: the meanings of the residue Ar were specified and tabulated down the equation.
For Schemes 90: the equations were modified and the meanings of Ar were inserted.
- Abbreviations must be explained at first appearance in the text, for the benefit of the reader. In conclusion, the material must be read carefully by the authors, and where appropriate, it is necessary to verify the information from the bibliographic source.
Was done